# Chemical, Physical, and Toxicological Properties of V-Agents

**DOI:** 10.3390/ijms24108600

**Published:** 2023-05-11

**Authors:** Georgios Pampalakis, Stavroula Kostoudi

**Affiliations:** Laboratory of Pharmacology, School of Pharmacy, Aristotle University of Thessaloniki, 54124 Thessaloniki, Greece; skostoud@pharm.auth.gr

**Keywords:** nerve agents, V-agents, phosphonothiolates, toxicity, physical properties, binary agents

## Abstract

V-agents are exceedingly toxic organophosphate nerve agents. The most widely known V-agents are the phosphonylated thiocholines VX and VR. Nonetheless, other V-subclasses have been synthesized. Here, a holistic overview of V-agents is provided, where these compounds have been categorized based on their structures to facilitate their study. A total of seven subclasses of V-agents have been identified, including phospho(n/r)ylated selenocholines and non-sulfur-containing agents, such as VP and EA-1576 (EA: Edgewood Arsenal). Certain V-agents have been designed through the conversion of phosphorylated pesticides to their respective phosphonylated analogs, such as EA-1576 derived from mevinphos. Further, this review provides a description of their production, physical properties, toxicity, and stability during storage. Importantly, V-agents constitute a percutaneous hazard, while their high stability ensures the contamination of the exposed area for weeks. The danger of V-agents was highlighted in the 1968 VX accident in Utah. Until now, VX has been used in limited cases of terrorist attacks and assassinations, but there is an increased concern about potential terrorist production and use. For this reason, studying the chemistry of VX and other less-studied V-agents is important to understand their properties and develop potential countermeasures.

## 1. Introduction

Although the Chemical Weapons Convention (CWC) was entered in 1997, chemical weapons still pose a threat and could be used by states or terrorists. The assassination of Kim Jong-Nam, the brother of Kim Jong-Un, with a binary version of VX, as well as the clandestine production of VX and sarin by Aum Shinrikyo are two examples that highlight their potential usage and associated hazards [1]. Nerve agents are among the most toxic and fast-acting chemical warfare agents. They are organophosphorus compounds grouped into four categories, the G-, the V-, the GV-, and the Novichok agents. The G-agents encompass tabun, its analogs and the esters of methylphosphonofluoridic acid such as sarin, soman, etc. These are volatile compounds mainly introduced through inhalation, although they can penetrate through the skin. Tabun was developed in Germany (where the G comes from) in 1936 by Gerhard Schrader. Later Schrader developed sarin (1938), while soman was developed by Richard Khun (Nobel laureate for his work on vitamins and carotenoids) in 1944. The V-agents constitute a group exhibiting higher toxicity than the G-agents. The letter V is derived from venomous, although victory, viscous and virulent have also been suggested as alternative sources [2]. They are oily liquids with low volatility that penetrate the skin easily; thus, they mainly constitute a percutaneous hazard. In 1952, the British chemist R. Ghosh, working in the Plant protection unit of Imperial Chemical Industries (ICI), synthesized the first V-agent, the Amiton (VG). Amiton was introduced in the market in 1954 but was found to be highly toxic for safe usage and it was withdrawn. Later, VX was identified, and its industrial production for military use was initiated in 1961. The GV-agents encompass molecules with structure R_2_N-P(O)(F)-OCH_2_CH_2_NR’_2_ (more complex compounds containing rings in the choline moiety, such as EA-5488, are also included in this group), and properties that are between the G- and the V-agents (Figure 1) [3]. The fourth group of organophosphate nerve agents is the Novichok agents, which were developed in Russia under the Foliant program. However, their chemical structure is still controversial, with some sources describing them as phospho(r/n)ofluoridate derivatives of substituted acetamide or guanidine [4], while others as organophosphate derivates of dihaloformadoximes [5]. Nonetheless, the first class has been considered the most likely and incorporated into the CWC list of banned chemicals. It should be noted that there is another class of nerve agents that are carbamates instead of organophosphates. These compounds contain quaternary nitrogen salts and are solids (an example is shown in Figure 1) [6].

Nerve agents gained publicity when sarin was illegally synthesized by the Japanese cult Aum Shinrikyo and used in the terrorist attacks in Matsumoto in 1994 with approximately 500 intoxications and in Tokyo Subway in 1995 with 6000 intoxications [7]. Sarin was also used during the Syrian civil war in 2013 and 2017 [8,9]. Recently, Novichok agents were used in a series of assassinations conducted at Salisbury and Amesbury in the United Kingdom [10]. A Novichok agent was also used on 20 August 2020 in an assassination attempt against the Russian activist Alexei Navalny during a domestic flight in Russia [11].

## 2. Methods

The literature search was performed in Pubmed and Pubchem (http://www.ncbi.nlm.nih.gov (accessed on 10 April 2023), in Google Patents, and in Google Scholar. Declassified documents on the physical properties and toxicities of various V-type nerve agents were also obtained from the Defense Technical Information Center (http://discover.dtic.mil (accessed on 10 April 2023). Keywords for searching were V-agents, VX, RVX, etc. Toxicities in some documents were reported as mm^3^·kg^−1^ and were converted to μg·kg^−1^ with the corresponding densities at 25 °C. Further, when toxicities were reported in μmol·kg^−1^, they were converted to μg·kg^−1^ using the corresponding molecular weight.

## 3. General Structures and (Sub)Categories of V-Agents

V-agents encompass a family of compounds that can be further classified into certain subclasses. In the present article, all the reported V-agents were classified according to their chemical structure. As shown in Figure 2, a total of seven V-agent subclasses were identified.

The most widely known and studied V-agents belong to class 1, which are phosphonylated thiocholines, such as VX and Russian VX (RVX or VR). Class 2 are the phosphorylated thiocholines (or phosphorylated analogs of class 1), such as the VG. Classes 3 and 4 encompass the phosphonylated and phosphorylated selenocholines, respectively, which are more toxic than their sulfur analogs; however, they are likely not of military importance since they are easily decomposed in the presence of air and release elemental selenium [12]. Class 5 includes V-agents that contain a thiocholine-like moiety (-SCH_2_CH_2_SCH_2_CH_3_), while V-agents belonging to classes 6 and 7 do not contain a choline moiety at all. Specifically, class 6 encompasses the 3-pyridyl phosphonates [13], and class 7, the 2-alkoxylcarbonyl-1-methylvinyl alkyl alkylphosphonates [14].

Figure 3 shows another class of highly toxic organophosphonate esters that have been patented [15]. Notably, these agents are salts and, therefore, solids that make their potential dispersion and application difficult, especially for potential terrorist attacks. Thus, this class of agents was not included in the original list in Figure 2.

Figure 4 shows similarities between certain V-agents and insecticides. From this, it can be deduced that certain V-agents have been designed based on insecticides by deleting the one -O- from the ester of phosphorus to convert it to phosphonate. For example, from systox (demeton), the V-sub x agent has been designed. The same is true for VE and the “parent” compound VG initially sold as a pesticide. Further, mevinphos is the parent compound of EA-1576 (Figure 4). As expected, and will be presented below, the introduction of the C-P bond increases the toxicity of the original pesticide compound. It should be noted that such an approach could be used in the future for the designing and clandestine development of new agents with high toxicity that are not covered by CWC.

*Stereochemistry.* Except for VG (and generally agents belonging to classes 2 and 4), all other V-type nerve agents display chirality and have two different isomers on the phosphorus atom. The isolated R_P_ and S_P_ isomers exhibit different toxicities that will be described later in this review. Other chiral centers or structural isomers may also be found in other molecules, e.g., in VP that carries the 3,3,5-trimethylcyclohexyl moiety. The weapons-grade VP is manufactured from the low melting 3,3,5-trimethylcyclohexanol (namely the *cis-* isomer). Regarding the different diastereomers over phosphorus atoms (e.g., VX, VR, etc.), the V-agents are always produced and used as racemic mixtures.

## 4. Physical Properties of V-Agents

The study of the V-agents’ physical properties is important since it correlates with their potential environmental spreading and contamination [16]. Generally, the V-agents are colorless to amber color oily liquids with low vapor pressure. Thus, they mainly constitute a percutaneous hazard in contrast to the G-agents. Their physical properties are summarized in Table 1 [17,18,19,20,21,22]. From these data, it can be deduced that increasing the number of carbon atoms in the phosphonothiolate molecule reduces the density except for EA-1622, EA-1694, and EA-1699.

A comparison of VE and VG indicates that the conversion of a phosphorate to a phosphonate also reduces the density of the compound. From the reported flash points, it is demonstrated that VS and EA-1728 exhibit higher flash points that correlate with an increased number of carbon atoms and increased molecular weight.

The viscosity decreases when the R group (directly attached to phosphorus) increases from methyl to ethyl, as shown for the pairs VX/VS, VM/VE, and EA-1699/EA-1694. On the other hand, when the R1 group increases, the viscosity increases, as shown for VX, compared to EA-1728 and EA-1763. Additionally, an increase in viscosity is observed as the number of carbon atoms in the R2 group (attached to nitrogen) increases, for example, VM compared to EA-1699. The flash points are between 135 and 170 °C for VE, EA-1622, VS, VX, and EA-1728.

Further, a comparison of VX with EA-1728 and EA-1763 shows that increased carbon atoms in the R1 group reduce the density. In general, the introduction of bulkier groups in R1 reduces the density, as shown with the EA-1521 agent that shows d = 0.995 g·mL^−1,^ but also with the *O*-cyclohexyl S-(diethylamino)ethyl methylphosphonothiolate that exhibits d = 0.9451 g·mL^−1^ @25 °C [21]. The introduction of the cyclopentyl group in R1 does not seem to reduce the density of the agent (agents EA-3148 and EA-3317 in Table 1). Finally, the lower consolute temperature of V-agents in water decreases with an increasing number of carbon atoms.

## 5. Toxicity of V-Agents

The V-agents, like all nerve agents, are potent irreversible inhibitors of the enzyme acetylcholinesterase (AChE) that hydrolyzes the neurotransmitter acetylcholine and is responsible for terminating the synapse transmission [2,23,24]. Due to their low vapor pressure, the V-agents mainly constitute percutaneous hazards, but inhalation of droplets during spraying is also a potential route of intoxication. The intravenous (*i.v.*) route of administration is a standard method for testing the toxicity of chemical warfare agents since it closely approximates the results obtained by inhalation; thus, it is practically important for field usage of agents [25].

Table 2 summarizes the toxicity of the agents in various experimental animal models and with various routes of administration. In general, V-agents exhibit lethality at the low μg·kg^−1^ levels. The toxicity data depicted in Table 2 [12,25,26,27,28,29,30,31,32,33,34,35] are obtained 24 h after exposure. For a faster time of action, increased dosages are required. For example, the 15 min LD50 for VE (EA-1517) administered *i.v.* in rabbits is 19.3 μg·kg^−1^ in comparison with 15.3 μg·kg^−1^ for 24 h. In this context, the 15 min LD50 for VP (EA-1511) administered *i.v.* in rabbits is 61.4 μg·kg^−1^ compared to 36.8 μg·kg^−1^ and for VG (EA-1508) is 135.9 μg·kg^−1^ compared to 52.2 μg·kg^−1^, respectively [25]. The 15 min time selected for toxicity studies is considered of military significance since it corresponds to rapid incapacitation. Similarly, when V-agents were applied on the bare skin of rabbits (*p.c.*) for 2 min or 24 h followed by calcium hypochlorite decontamination and the animals were further monitored for 24 h, the LD50s increased for EA-1511 from 81.8 μg·kg^−1^ for 24 h exposure to 910.5 μg·kg^−1^ for 2 min exposure and for VE, from 40.7 μg·kg^−1^ for 24 h exposure to 6413.4 μg·kg^−1^ for 2 min. [31].

As mentioned, the V-type nerve agents are composed of enantiomers at phosphorus atom except for Classes 2 and 4 due to the presence of four different substituents. The S_P_ isomers show more potent AChE inhibition in vitro, and exhibit increased toxicity in vivo. For example, the racemic VX exhibits *s.c.* LD50 in albino rats 13.1 μg·kg^−1^ while the S_P_-(-)-VX has 8.8 μg·kg^−1^ and the R_P_-(+)-VX has 56.1 μg·kg^−1^ [28].

*Connection of structure with toxicity.* Significantly increasing the size of the R group directly attached to the phosphorus, for example, by substituting the methyl group with a phenyl group, results in a significant reduction of toxicity. In this direction, the phenyl-VX analog (PhX) of the VX is a slowly reversible inhibitor of *Electrophorus electricus* AChE in contrast to irreversible inhibition exerted by V-agents. The IC_50_ against AChE for 2 h reaction for the PhX is 20 ± 2 nM while for VX, only 0.3 ± 0.05 nM [36].

Replacement of sulfur from the thiocholine group in classes 1 and 2 agents with selenium increases toxicity. For example, the *s.c.* LD50 toxicity of VG in mice is 190 μg·kg^−1^ [35], while for the seleno-VG is 60 μg·kg^−1^ [12]. On the other hand, the replacement of sulfur with oxygen results in the generation of reversible inhibitors [37,38] that exhibit significantly reduced toxicity, as demonstrated with the *O*-analog (choline) of VX [i.v. LD50 204,000 μg·kg^−1^ and 164,000 μg·kg^−1^ in mice and rabbits, respectively (PUBCHEM, https://pubchem.ncbi.nlm.nih.gov/compound/51437#section=Acute-Effects (accessed on 3 March 2023))]. In the VG-type agents, bridging the two alkoxy groups, such as the compound shown in Figure 5, results in the elimination of their toxicity. Specifically, the cyclic phosphorous compound shown in Figure 5 exhibits a *s.c.* LD50 in mice 700,000 μg·kg^−1^ [39] compared to VG, that is 190 μg·kg^−1^ for the same route of administration as previously mentioned.

Reducing the basicity of the compound reduces binding to AChE and reduces toxicity, as demonstrated for the fluorinated substitutes of VG (Figure 6A) [40]. As shown in Figure 4, deletion of one -O- from a phosphoro-molecule and conversion to phosphonate results in significantly increased toxicity with potential chemical warfare ability. Nonetheless, deletion of both -O- and conversion to phosphinate does not affect toxicity, as shown in Figure 6B [41].

Quantitative structure-activity relationship (QSAR) has been applied to select potential insecticides with lower toxicity but can be applied to generated compounds that exhibit increased toxicity [41]. In the same direction, drug discovery artificial intelligence platforms can be repurposed to identify new toxic chemical warfare agents. Indeed, this has been performed, and compounds expected to have significantly higher toxicity than VX were designed. Nonetheless, their chemical formulas have not been disclosed [42]. However, this provides an example of how new V-type agents can be designed that pose additional threats due to possible resistance in therapeutic regimens or persistence in the environment, etc.

*Effect of penetration enhancers.* The percutaneous toxicity of V-agents can be further increased through the co-application with penetration enhancers. Dimethyl sulfoxide (DMSO) is one of the penetration enhancers that has been investigated. As shown in Table 2, the percutaneous toxicity of EA-1728 is 59.1 μg·kg^−1^, but it falls to 10.1 μg·kg^−1^ using DMSO [32]. Other penetration enhancers that have been tested are n-octylamine, n-decylamine, and n-dodecylamine. Specifically, these agents enhance the percutaneous effectiveness of VX in human volunteers and reduce the dose required to develop symptoms [43].

*Aging.* Nerve agents react with the serine hydrolase AChE by phosphonylating (or phosphorylating for Classes 2 and 4) the active site serine. The adduct is amenable to a dealkylating reaction known as aging [43,44,45]. The rate of aging is of pharmacological significance since the “aged” enzyme is resistant to oxime reactivation [the currently used drugs against nerve agent poisoning such as pralidoxime chloride (2-PAM)]. The rate of aging depends on the chemical structure of the alkoxy group. Aging takes place through cleavage of the O-C bond; therefore, substituents that stabilize the formation of carbocations result in rapid aging, as observed in soman that carries the pinacolyl group (Figure 7) [43,45]. Soman has been reported to cause aging within 0.07 h, while VX, VR, and CVX age after 36.5, 138.6, and 32.2 h, respectively [46]. It is expected that agents such as VT with pinacolyl groups or other groups that can form stable carbocations that will cause rapid aging; thus, they will be resistant to oxime reactivation approaches. Further, rapid aging has been observed for the 2,2-dimethylcyclohexyl group, which is analogous to pinacolyl [45]. Interestingly, VR-56 has also been reported to exhibit rapid aging, although the mechanism is not reported [41].

*Non-AChE targets of nerve agents.* Unexpectedly, when the *Ache*^−/−^ mice were generated, they were born and could survive for 14 days. Notably, subjecting the animals to high-fat diet increased life expectancy for over 1 year [47]. This indicates that other alternative mechanisms can partially compensate for the absence of AChE. The *s.c.* LD50 of VX is 24 μg·kg^−1^ for wt mice, 17 μg·kg^−1^ for the heterozygous *Ache*^+/−^ mice, and 10–12 μg·kg^−1^ for the *Ache*^−/−^ mice [48]. This demonstrates that nerve agents bind to other biological targets to exhibit lethality. One of these targets could be butyrylcholinesterase (BuChE), but others remain to be identified. Accordingly, the *Ache*^−/−^ mice are an important in vivo “tool” to fish potential non-AChE targets.

Neuropathy target esterase (NTE) is another potential biological target of nerve agents. Indeed, NTE is inhibited by certain organophosphate agents with a major representative, the tri-o-cresyl phosphate (TOCP). NTE is responsible for organophosphate-induced delayed neuropathy (OPIDN) due to axonal degeneration in the peripheral and central nervous system. OPIDN begins with mild symptoms, such as weakness, ataxia, muscle twitching in the legs, and tingling [49,50]. Finally, it leads to flaccid paralysis starting from the toes and continuing to the hands and thighs [49]. The term “delayed” is used since there is a clinically quiescent period after exposure to organophosphate and the appearance of symptoms. This period may be between 4 days to 4 weeks. Importantly, smaller than lethal doses of organophosphates may inhibit NTE and cause OPIDN. For example, after the terrorist attack with sarin in Tokyo subway in 1995, certain individuals suffered from OPIDN [51].

It should be mentioned that rodents, although widely used as animal models for nerve agent intoxication, are resistant to OPIDN and thus cannot be used for OPIDN studies [52]. Alternative animal models are required, such as cats, dogs, or chickens. The latter model is considered to have equal sensitivity of humans for OPIDN and is approved by the US EPA for screening for OPIDN. Using chickens, it was shown that VX does not cause OPIDN, consistent with the in vitro finding that VX is 1000-fold less active in inhibiting NTE than sarin [53]. Morover, VR-56 does not cause OPIDN [41]. Whether there are other V-agents that can cause OPIDN has not been examined in detail and merits future investigation as it will assist in designing new medical countermeasures.

## 6. Exposure to V-Agents

V-type nerve agents have not been used in military conflicts or extended terrorist attacks. However, VX has been associated with the Utah accident in 1968 and has been used as an assassination weapon in a limited number of cases. Further, the pathological effects of VX and EA-3148 have been tested at low levels in human volunteers [43,54,55].

*Utah accident.* In 1968 in Dugway Proving Ground in Utah, during open-air field testing, a malfunction in the chemical dispersion system attached to the aircraft carrying two tanks filled with VX resulted in the release of VX at a higher altitude. This, combined with changes in weather conditions, resulted in the transportation of the agent approximately 45 km away from the field site to a valley where sheep were grazing, causing the death of more than 6000 sheep [56].

*VX as an assassination weapon and effects of VX intoxication.* The widely known case of assassination with VX is of Kim Yong-Nam, the brother of Kim Yong-Un, in 2017, at Kuala Lumpur International Airport in Malaysia [1]. In this case, it is speculated that a new form of binary VX was used, as will be outlined in the next section.

There are additionally three cases of VX intoxication in humans. Specifically, VX was used by the Japanese cult Aum Shinrikyo in 1994 against two persons (on 2 and 12 of December) and in 1995 (on 4 January) against one person. The victim of the VX attack on 12 December 1994 died while the other two survived. Death was attributed to VX 7 months after exposure when VX-metabolites were finally identified [57,58]. The symptoms of one of surviving persons have been recorded in detail. VX caused impaired vision, seizures, and loss of consciousness. When the patient was administered to the hospital, they were found to exhibit frothing at the mouth, sweating, and cyanosis. Further, fasciculations, high blood pressure and hypothermia were found. The patient received 3 mg atropine per day *i.v.* and was supported by mechanical ventilation. The serum cholinesterase levels were normal 20 days after intoxication, while 6 months later, the patient suffered from antegrade and retrograde amnesia. In contrast to sarin, which evokes mainly nicotinic responses, VX evolves mainly muscarinic responses [59].

*Exposure of volunteers to VX.* There has been a clinical study where seven individuals were given *i.v.* injections of the nerve agent VX. A single dose of 1 μg·kg^−1^ resulted in the fall of RBC cholinesterase to levels 40–50%, while 2 μg·kg^−1^ is considered a toxic dose that requires medical assistance, including administration of atropine and oximes [43].

*Low-level exposure of EA-3148 in humans.* The EA-3148 has been tested in volunteers to shed light on its pathophysiological effects. It is a more potent cholinesterase inhibitor than VX and exhibits higher toxicity (Table 1). When a dose of 1.15 μg·kg^−1^ was administered *i.v.* in humans, it exhibited a rapid reduction of RBC cholinesterase within 15 min. Two out of thirteen volunteers exhibited toxic signs 5 to 8 min following exposure that included sweating, dizziness, weakness, and tiredness. Other symptoms of exposure were anorexia, poor sleep, fatigue, blurred vision, salivation, etc., as expected for nerve agent exposure [56].

## 7. Stability of V-Agents during Storage

Class 1 V-agents are subjected to gradual deterioration upon storage due to various impurities derived during the process of synthesis. These impurities include thiolamine, N,N-diisopropylaminoethanethiol, p-diethyldimethylpyrophosphonate (referred to as pyro), ethyl methylphosphonic acid, alcohols, and water [60,61]. As expected, the rate of agent decomposition increases with temperature and in the presence of rust. The main route of decomposition is hydrolysis, which is autocatalytic and depends on the presence of pyro impurity (Figure 8) [62]. Pyro reacts with water to yield phosphonic acid that reacts with VX yielding more pyro. Thus, pyro is harmful only in the presence of water. The importance of the amino group that should be protonated in the autocatalytic mechanism is demonstrated by the fact that the compound CH_3_P(O)(OC_2_H_5_)(SC_2_H_5_) does not display an autocatalytic decomposition process [62]. The production of polymers during the degradation of the agent, as shown in Figure 8, is partly responsible for the high viscosity exhibited by the final degradation mixture.

To increase the stability and thus the self-life, it is necessary to add compounds that inhibit decomposition. Soluble carbodiimides have been used for the stabilization of VX. Carbodiimides react with water, alcohols, and thiols, and the products do not further react with VX. Additionally, the reaction of carbodiimides does not evolve gases that will increase the pressure inside the containers. The addition of 2% N,N′-dicyclohexylcarbodiimide in VX reduces its decomposition from 10% in two months and 20% in 3.3 months (neat agent) to 10% in 3.3 months and 20% in seven months, respectively, at 71 °C. Increasing the carbodiimide to 6% further increases the stability of VX, with 10% decomposition occurring after nine months at 71 °C. Thus 2–4% carbodiimides ensures the stability and storage of the agent for several years at ambient temperature [63]. Except for N,N′-dicyclohexylcarbodiimide, other carbodiimides, such as diisopropylcarbodiimide, di-*O*-tolylcarbodiimide, etc., can alternatively be used. In the presence of carbodiimides, the biological potency of the percutaneous application of VX is not altered [63].

In terms of stability, the classes 3 and 4 have also been investigated. These seleno-derivatives of the V-agents are likely of no potential military or terrorist use due to instability in the presence of oxygen. Further, the intermediates during synthesis are unstable in the presence of oxygen and react to produce elemental selenium [12].

## 8. Unitary vs. Binary Formulations and Thickening of V-Agents

The V-agents are disseminated as liquid droplets that contaminate equipment, soils, buildings, vegetation, etc. Weapons containing the toxic agent are known as unitary weapons. The high toxicity of the nerve agents posed problems in safe handling, and storage. This led to the development of binary weapons. These weapons contain two relatively non-toxic and easy-to-handle compounds that mix to produce the active agent after launching. The most classical weapon developed was the BLU-80/B BIGEYE, designed to deliver binary VX (VX2). This is a 500-pound glide bomb that, during flight, generates VX from the reaction between *O*-diisopropylaminoethyl O′-ethyl methylphosphonite (QL) and sulfur (Figure 9) and then disperses the agent in the designated area. QL and sulfur are kept in separate containers and assembled before the flight. QL is a viscous liquid with low volatility, a high flash point (170 °C), and a fishy odor. It is not corrosive and has low toxicity; thus, it is safe for transportation and handling. Except with sulfur, it can also react with selenium, but the product has not been described [64], although it can be assumed that it will yield the seleno-VX.

*Could a new binary VX have been used to kill Kim Jong-Nam?* As mentioned, VX was used in the assassination of Kim Jong-Nam at Kuala Lumpur Airport. Specifically, two women sequentially rubbed their hands in Nam’s face. In the shirt of the first woman, ethyl methylphosphonic acid was detected [MeP(O)(OEt)OH], while from the T-shirt and the fingernails of the second woman, the following compounds were detected: VX, 2-(diisopropylamino)ethyl chloride, 2-(diisopropylamino)ethanethiol and bis(2-diisopropylaminoethyl) disulfide (Table 3) [1,57]. The identification of different chemicals in these two persons indicates that VX was used in a binary form and was generated after the second woman rubbed her hand on Kim’s face. VX probably entered the body through the eyes since Kim died 20 min after exposure. If VX had entered through the skin, it would have required more time for intoxication.

Interestingly, the exact formula of the binary VX used to intoxicate Kim has not been identified. Direct reaction of ethyl methylphosphonic acid with 2-(diisopropylamino)ethanethiol at room temperature cannot take place unless there is a catalyst that was not detected (Figure 10). In addition, under this reaction scheme, the role of the 2-(diisopropylamino)ethyl chloride is unknown. Regarding bis(2-diisopropylaminoethyl) disulfide, this is probably derived from air oxidation of the 2-(diisopropylamino)ethanethiol, while the bis(2-diisopropylaminoethyl) sulfide from the reaction between 2-(diisopropylamino)ethanethiol and 2-(diisopropylamino)ethyl chloride. Finally, there is no clue on how the 2-(dimethylamino)ethanol was found in the samples obtained from Kim’s body.

*Thickening.* It is the mixing of a nerve agent with a polymer with the aim of increasing persistency and adherence to surfaces. For example, VX can be thickened in the presence of a thickener, e.g., 1% *w*/*w* polyisobutyl methacrylate and 2% *w*/*w* multicellular polyethylene. The materials are mixed for 2 to 4 h to thoroughly disperse all compounds under a nitrogen atmosphere. The thickened agent is then delivered as the neat agent [65]. Polymethyl methacrylate can also be applied for V-agent thickening [66].

## 9. Chemical Synthesis

The method(s) for industrial synthesis is/are important given that there was a case where a terrorist group had developed a factory for large-scale clandestine production of nerve agents.

### 9.1. Major Industrial Production Process

VX is the most common nerve agent, and its most classical industrial preparation process is based on the reaction between QL and sulfur, which are also the components of the binary VX. QL is synthesized in three steps. The first step is the production of the methylphosphonous dichloride (SW) from phosphorus trichloride and methane. The next step is the conversion of SW to diethyl methylphosphonite (TR), and the final step is the conversion of TR to QL.

The conversion of PCl_3_ and CH_4_ to methylphosphonous dichloride is carried out in the presence of a small amount of O_2_ at 500 °C. This reaction has been studied in detail, and a few oxygenated byproducts are also generated, as shown in Figure 11 and Figure 12 [67]. Except for methane, the reaction can be carried out with natural gas treated to remove the small amount of higher-than-ethane alkanes. Further, PCl_3_ reacts with ethane under the same conditions to produce ethylphosphonous dichloride, a precursor of VE and VS but also a potential precursor for the pesticide fonofos.

For the conversion of SW to TR, SW, and ethanol are introduced in the reactor in the presence of NH_3_ (to neutralize the HCl byproduct) as well as isobutane, which acts as an auto-refrigerant and diluent. After passing the mixture from the main reactors, NaOH in water is added to remove ammonium chloride and other water-soluble byproducts.

In the last step, streams of TR and KB are mixed, and steam heated in the presence of glacial acetic acid to catalyze the transesterification reaction by neutralizing the basic impurities present in the TR. The byproduct EtOH is removed by distillation and recovered. After this step, the mixture contains QL, LT [bis(2-diisopropylaminoethyl) methylphosphonite], and unreactive TR. This mixture passes through the TR recovery and flash still column, where TR is removed and recycled for reuse, and QL is purified and eventually stored [64].

Small-scale production of QL in the laboratory has been described in a patent, where the reaction between SW and ethanol is carried out in dried diethyl ether, and the transesterification is conducted under reflux with continuing removal of ethanol [68].

### 9.2. The EMPTA Process

In this process, the EMPTA (*O*-ethyl methylphosphonothioic acid) constitutes the major intermediate [68]. The process starts as previously, with the reaction of methane with phosphorus trichloride. Then, the SW reacts with sulfur to produce methylphosphonothioic dichloride CH_3_P(S)Cl_2,_ which is further converted to EMPTA through a reaction with ethanol and KOH. EMPTA is converted to VX through a reaction with diisopropylaminoethyl chloride (Figure 13). It is expected that changing the substituents in the reactants will result in the production of other V-type nerve agents. The detection of EMPTA in soil samples was used as an indicator for the alleged development of VX in the Sudanese pharmaceutical industry Al Shifa, which was destroyed by a missile attack [69].

A patent that describes in detail the conditions to generate V-type nerve agents from EMPTA has been disclosed [70]. In accordance, the *O*-ethyl ethylphosphonothioic acid can be used to produce VE and VS. Indeed, the aforementioned patent also describes the generation of other V-agents from EMPTA analogs, e.g., the isopropylphosphonothioic salt thus providing proof that such modification can take place for the preparation of novel class 1 V-agents. Notably, these reactions are carried out in aqueous solutions, making them easier to perform.

Finally, the same patent discloses a potential new type of binary agent that can be formed in situ to contaminate water supplies. Specifically, it involves adding the hydrochloride salt in a water supply of β-chloroethyl diethylamine, NaOH, followed by sodium ethyl methylphosphonothiolate [70].

### 9.3. Production Process Based on O,O-Dialkyl Alkylphophonothionate Starting Material

Another method to produce V-agents has been described in another patent with VE as an example [71]. The process starts with half-hydrolysis of *O,O-*diethyl ethylphosphonothionate by refluxing in the presence of ethanolic sodium or potassium hydroxide. The resulting acid (*O*-ethyl hydrogen ethylphosphonothionate) is purified by removal of methanol, dilution of the residue in acidified water, extraction with ether, and distillation. The acid is allowed to react with sodium methoxide to yield the sodium salt in methanol. Methanol is then removed azeotropically with benzene, and the salt is treated with diethylaminoethyl chloride under reflux. Then, petroleum ether is added, and the byproduct NaCl is removed with filtration. VE is purified from filtrate with distillation.

The importance of this procedure lies in the fact that it can be carried out in water with a small modification [71]. The first step, that is, the production of the acid, is the same as described. Subsequently, the acid is converted to salt with aqueous NaOH. An equimolar mixture of NaOH and hydrochloride salt of diethylaminoethyl chloride is added. The reaction is allowed to stand for several hours and extracted with benzene. The product is purified by distillation. This modification is called the “water method”. The water method can be used to prepare V-agents with the -OR group being ethyl or isopropyl, R-P being ethyl or methyl, and R1 ethyl. If the R1 is methyl, then dimerization of diethylaminoethyl chloride to tetramethyl piperazinium compound occurs.

### 9.4. Synthesis of EA-1576

The EA-1576 has *cis* (*Z*) and *trans* (*E*) isomers. The *trans* isomer exhibits increased percutaneous and intravenous toxicities (although no values have been disclosed) [14]. A patent describing the identification and production of EA-1576 has been disclosed. Two processes have been described: the amine method and the alkali method. In the amine method, the alkylphosphonic dichloride reacts with the appropriate alcohol, e.g., 3-methylcyclohexanol. In the amine method, the produced alkyl alkylphosphonic chloride reacts with the alkyl acetoacetate (ethyl for EA-1576) in the presence of an amine, usually NEt_3_ to neutralize the HCl byproduct at low temperature (e.g., 35 °C). In the alkali method, the sodium salt of the acetoacetate reacts with the alkyl alkylphosphonic chloride at high temperatures (e.g., 80 °C). The alkali method yields preferentially the *trans* isomer while the amine is the *cis* isomer. The *cis* isomer can be converted to *trans* by heating it at 100 °C in 4% sodium ethyl (or methyl) acetoacetate with 90% yield. Further, long-term storage of the *cis* isomer at 50 °C converts it to the *trans*.

For example, for the synthesis of the EA-1576 *trans* isomer, methylphosphonic dichloride reacts with methylcyclohexanol in the presence of NEt_3_ in benzene while cooling. Then, new NEt_3_ is added along with ethyl acetoacetate with a temperature maintained up to 20 °C. After completion of the reaction (2 h), the reaction is washed with 5% NaOH, extracted, and the solvent removed by distillation at reduced pressure. The product is then isomerized with the addition of 4% sodium salt of methyl acetoacetate and heated at 100 °C for 1 h (Figure 14).

### 9.5. Small-Scale Synthesis for Specific Purposes

The synthetics schemes provided in this paragraph have been separated from the previous ones since they are to be used in special cases and specifically for the preparation of pure enantiomers or radiolabeled compounds for biological experiments. However, they can readily be adaptive for the synthesis of relatively larger quantities of V-agents.

*Synthesis of purified enantiomers.* The preparation of pure enantiomers is mainly a research focus of academic labs. Thus, there are no industrial processes for their synthesis. The synthesis of the pure enantiomeric V-agents is based on the phosphonylation of l-(-)-ephedrine, followed by the separation of produced diastereomers. Specifically, l-(-)-ephedrine reacts with methylphosphonothioic dichloride, and the resulting diastereomers are separated with preparative thin-layer chromatography using hexane/acetone (4:1) as a solvent mixture. Then, the isolated diastereomers are esterified and hydrogenated to remove the l-(-)-ephedrine. Finally, the phosphonate reacts with the appropriate choline chloride to yield pure enantiomers [72]. The schematic synthesis is illustrated in Figure 15.

*Synthesis of radiolabeled V-agents*. The radiolabeled V-agents are to be used in vivo studies, e.g., for the determination of percutaneous penetration rate, and generally track the V-agent when administered in a laboratory animal. The synthesis scheme shown in Figure 16 has been applied for the synthesis of ^14^C-radiolabeled VX and VM with an overall yield of 40 and 41%, respectively [29]. This method can be adjusted accordingly to generate any phosphonothiolated V-agent.

*Tracing of the synthetic route*. In the case of the use of chemical warfare agents during armed conflicts between countries, there is no point in developing methods to identify the perpetrator since it is probably already known from the beginning. Nonetheless, it is very important to identify the perpetrator in asymmetrical attacks and terrorist acts. For this, tracing the synthetic route through the chemical attribution signatures is needed. These signatures include the impurities and byproducts from the different reaction schemes [73]. The application of the methodology to identify the synthetic route used to produce VR that was spiked in various food samples has been demonstrated [74].

## 10. V-Agent Surrogates and Mimics

Due to the high toxicity of nerve agents and the associated difficulties when handling these chemicals, nerve agent surrogates and mimics have been developed. The surrogates constitute agents that react with AChE and form adducts that resemble the adducts formed by the reaction of the nerve agent with the AChE. These molecules are important in the development of new therapeutic approaches that aim to reactivate AChE [75]. The mimics are compounds that chemically react similarly to the parent agent but exhibit significantly less toxicity and can be applied in testing sensing devices or in testing the mechanism of sorption and contamination of various surfaces by the nerve agents [76,77].

*Surrogates.* The most common surrogates are the phospho(n/r)ylated 4-nitrophenyl esters. The 4-nitrophenyl group constitutes the leaving group that allows the formation of the adduct. The other moieties attached to the phosphorus will generate an identical nerve agent under study enzyme–adduct. For example, the 4-nitrophenyl ethyl methylphosphonate (NEMP) is the classical VX surrogate that will generate the *O*-ethyl methylphosphonate-AChE adduct that is identical to the adduct formed by the reaction between VX and AChE [78]. The 4-nitrophenyl isopropyl methylphosphonate (NIMP) and the phthalimidyl *O*-isopropyl methylphosphonate (PIMP) are the sarin surrogates [78] but can also be applied as surrogates for the EA-1728. Their synthesis is simple and can be carried out with methylphosphonic dichloride as the starting material [78] or from bis-nitrophenyl methylphosphonate [79], as shown in Figure 17.

*Mimics.* Classic examples of nerve agent mimics are diethylchlorophosphate and diisopropyl fluorophosphate (DFP), used as a mimic for sarin [80]. For V-agents, the *O*-analog of VX (VO) and the insecticide methyl-demeton have been used [76]. Based on this study, we assume that the *O*-analogs of V-agents that are significantly less toxic, as mentioned previously and thus safer to use, are promising compounds as V-agent mimics. Another V-agent mimic is the diethyl methylthiomethylphosphonate [(C_2_H_5_O)_2_P(O)CH_2_SCH_3_] [81]. Sometimes there is confusion between mimics and surrogates; the term surrogate can include mimics [77]. There are also cases where a compound can be both a surrogate and a mimic, such as the already mentioned CH_3_P(O)(OC_2_H_5_)(SC_2_H_5_) that can react with AChE forming a VX-AChE adduct but also can be applied as a mimic for V-type nerve agents.

## 11. V-Agent Antidotes

The treatment of nerve agent poisoning is based on the reactivation of AChE with oximes (Figure 18) with concurrent administration of atropine or a related anticholinergic compound [82,83]. Nevertheless, it is known that better treatment is achieved when reversible AChE inhibitors have been given prior to nerve agent exposure as a prophylactic approach. Currently, pyridostigmine, a carbamate drug used in myasthenia gravis, is distributed to military personnel to be given as a pretreatment in case of a potential nerve agent exposure. Pyridostigmine works by blocking the nerve agents to inactivate AChE through competition [84]. Another prophylactic treatment involves the administration of plasma-derived BuChE or recombinant BuChE that acts as a bioscavenger. In this direction, a pegylated recombinant human BuChE produced in transgenic goats and named Protexia^TM^ has been developed. Importantly, Protexia^TM^ can be administered either pre- or post-exposure [85]. The therapeutic efficacy of Protexia has been tested in animals. For example, Protexia^TM^ was effective in successfully rescuing all guinea pigs challenged with VX applied *p.c.* at 2.5-fold LD50 when administered 2 h after exposure, in contrast to the control, where all animals succumbed [86].

As mentioned, to reactivate inactivated AChE, oximes must be administered [87]. The history of the development of oxime reactivators has been reviewed elsewhere [88]. The two systems containing oximes used by the US Army include the Antidote Treatment Nerve Agent Auto-injector (ATNAA) and the newer Improved Nerve Agent Treatment System (INATS) [89]. The ATNAA is a single-dose autoinjector that contains atropine and the chloride salt of 2-PAM [2] (Figure 19).

2-PAM is effective in the treatment of VX poisoning but not VR. This is based on the fact that, except for the aging of phospho(n/r)ylated AChE that accounts for resistance to reactivation and was analyzed previously within this article, there are also steric effects that account for resistance to oxime-based reactivation of phosphor(n/r)ylated AChE. When the molecular volume of the alkoxy group attached to the phosphor(n/r)ylated AChE increases beyond isopropyl, e.g., VR (isobutyl), etc., then 2-PAM cannot be positioned to the choline-binding site of AChE to exert its reactivating ability. On the other hand, bis-pyridinium oximes bind to the peripheral site of AChE and can reactivate the enzyme even when the phosphonylated AChE contains larger RO- groups [90]. Thus, the newer improved antidote for nerve agents, the INATS, contains a bis-pyridinium oxime, the MMB-4. Further, atropine has been substituted with scopolamine [89] (Figure 19).

Overall, the current nerve agent treatment protocol involves the administration of pyridostigmine (if nerve agent exposure is suspected; in case of a terrorist attack, this will be highly unlikely), atropine, oxime, and a benzodiazepine (diazepam or midazolam) to fight the seizures [91]. Future research aims to fight V-agent, and generally, nerve agent exposure will include the designing of new oximes. In this endeavor, SAR studies will be helpful [92], as well as the designing of novel detoxifying enzymes. In the latter direction, an engineered phosphotriesterase to specifically detoxify the VX, VR, and CVX has been developed [93]. Finally, assessment of oxime-based therapeutics is necessary for the less studied VP and EA-1576 agents.

## 12. Conclusions

Previous reviews in the field of nerve agents have dealt with the therapeutic strategies used as medical countermeasures with a special focus on oximes [88,94]. Thus, in the present review, only a brief discussion on antidotes against V-agent poisoning is provided. Other reviews have also focused on the destruction and decontamination of chemical warfare agents [60]. Thus, these sections are also not covered in this article. Instead, this article provides a detailed analysis and presentation of all currently reported V-agents that go beyond that classically studied VR and VX. For the first time, these agents have been classified into subgroups according to their formulas. This classification system has highlighted the potential classes of yet unexplored V-agents that could be used in terrorist attacks. This is also essential since their methods of manufacturing are now readily available in a detailed manner in various documents that facilitate their illicit development. This is important to increase awareness and drive the development of countermeasures other than the VX and VR agents. Indeed, the development of potential AChE reactivators is tested against the standard VX and VR agents with unknown pharmacological efficacy against the other classes of V-agents. Further, it highlights the need to expand the CWC to include these new V-agents. Nevertheless, it is interesting to note that artificial intelligence platforms used in drug discovery can be repurposed to identify new toxic chemical warfare agents [42]. Although in the reported study [42], new chemical agents expected to exhibit significantly higher toxicity than VX were predicted, and their formulas were not revealed, these studies may be replicated by other researchers with a focus on designing and developing new agents.

## Figures and Tables

**Figure 1 ijms-24-08600-f001:**
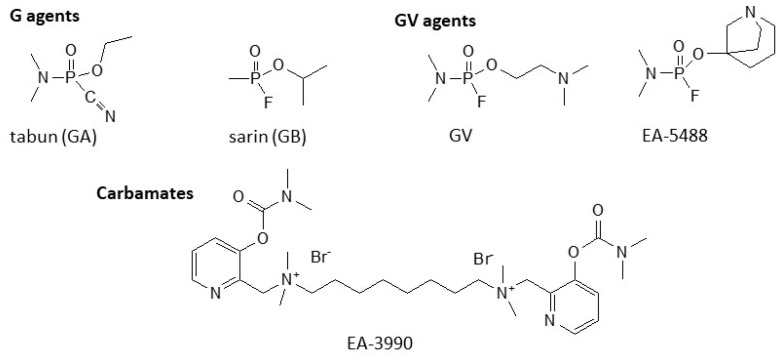
Examples of G-, GV-, and carbamate nerve agents.

**Figure 2 ijms-24-08600-f002:**
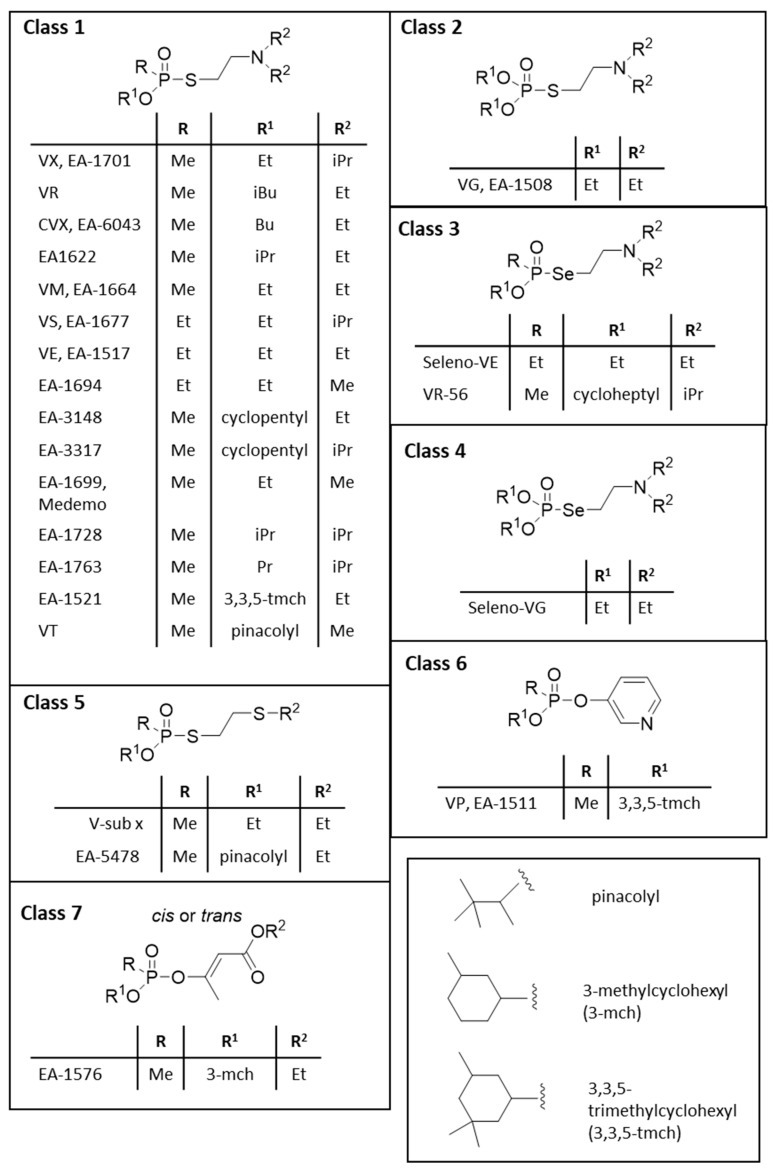
Classification of V-agents according to their chemical structure. Seven different classes can be identified named classes 1–7.

**Figure 3 ijms-24-08600-f003:**
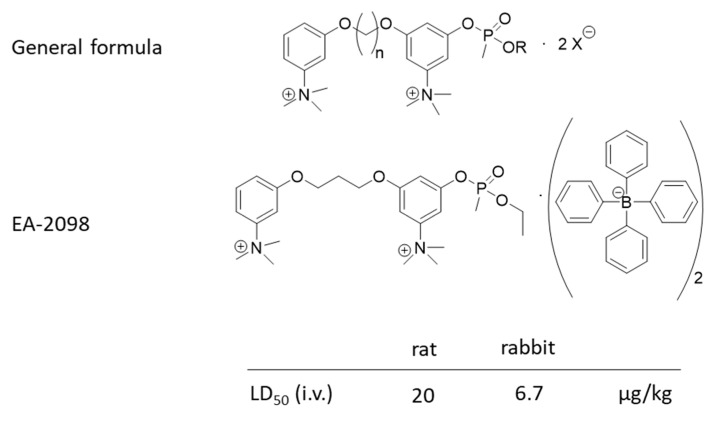
General formula of a class of highly toxic organophosphate salts and the structure and toxicity of the most toxic representative compound (EA-2098).

**Figure 4 ijms-24-08600-f004:**
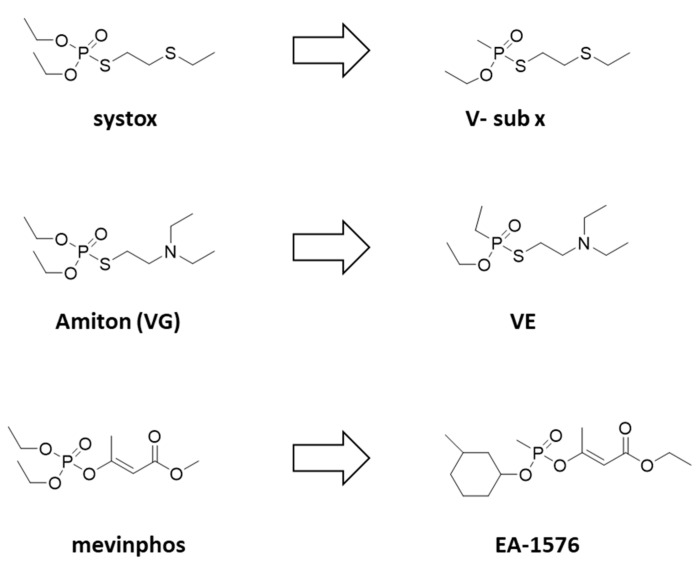
Chemical connection between certain V-agents and insecticides.

**Figure 5 ijms-24-08600-f005:**
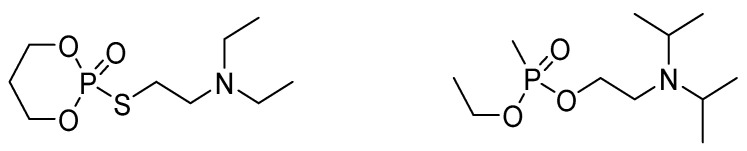
(**Left**)A bridged VG agent. Bridging the R groups results in a loss of toxicity. (**Right**) the *O*-analog (choline) of VX, also referred as VO.

**Figure 6 ijms-24-08600-f006:**
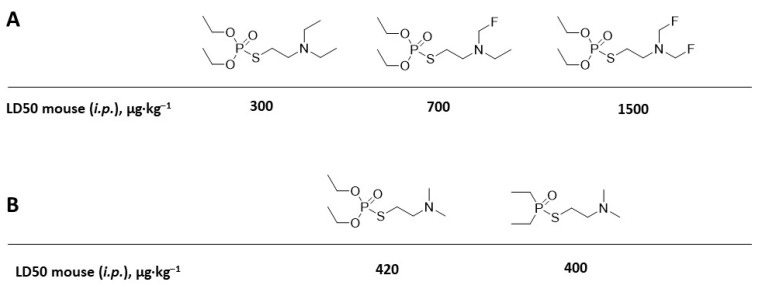
(**A**) Effect of reduction in basicity on the toxicity of the nerve agents. (**B**) Effect of converting a phosphorate to phosphinate.

**Figure 7 ijms-24-08600-f007:**
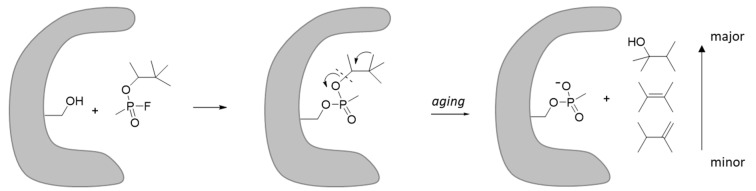
Example of the reaction between soman with AChE followed by aging. Minor and major indicate the byproducts of the aging reaction.

**Figure 8 ijms-24-08600-f008:**
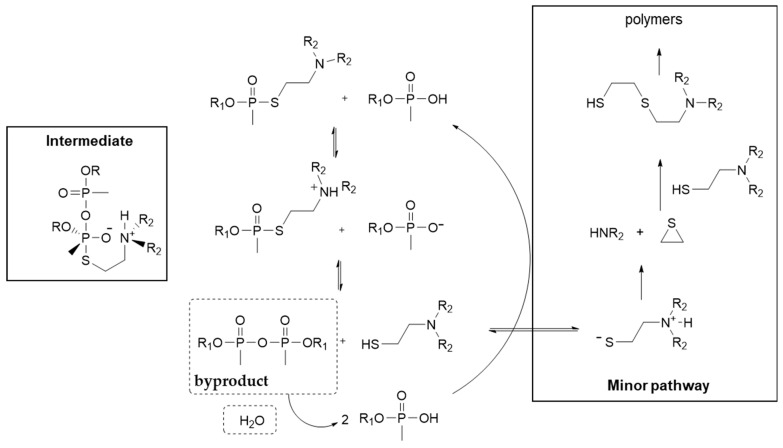
Mechanism of autocatalytic hydrolysis of V-type nerve agents (VX and VR have been tested). The pyro and water necessary for the autocatalytic cycle are shown inside dashed boxes. Removal of water stabilizes the nerve agent. The polymers based on thiol reactions shown on the right are responsible for the observed high viscosity of the final reaction mixture. A theoretical structure of the intermediate that results in the generation of pyro and thiol, which is also the rate-determining step, is shown on the left. VX reacts slower than VR due to the presence of the sterically hindered isopropyl groups.

**Figure 9 ijms-24-08600-f009:**
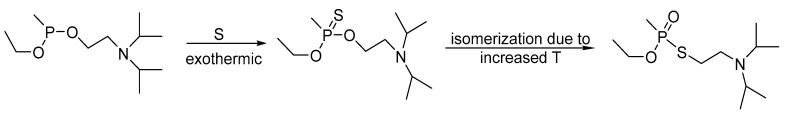
Binary VX.

**Figure 10 ijms-24-08600-f010:**
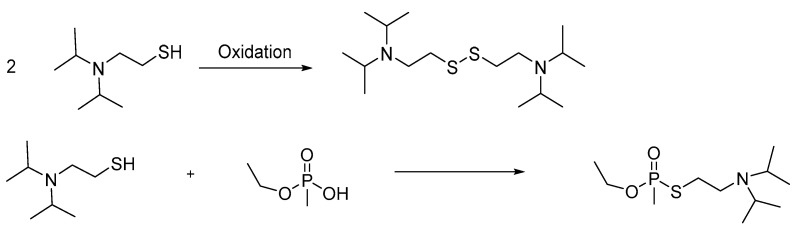
The presence of bis(2-diisopropylaminoethyl) disulfide can be attributed to the oxidation of 2-(diisopropylamino)ethanethiol. However, the reaction of ethyl methylphosphonic acid with 2-(diisopropylamino)ethanethiol cannot occur at room temperature. Thus, the exact mechanism of VX formation and whether a catalyst was used is yet unknown.

**Figure 11 ijms-24-08600-f011:**
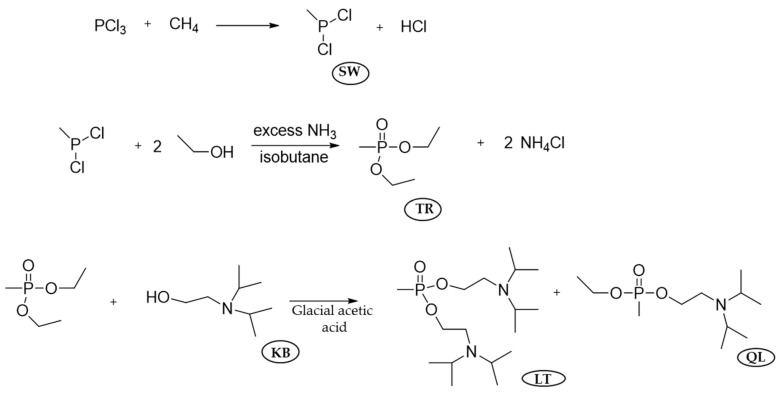
The major industrial production process of QL.

**Figure 12 ijms-24-08600-f012:**
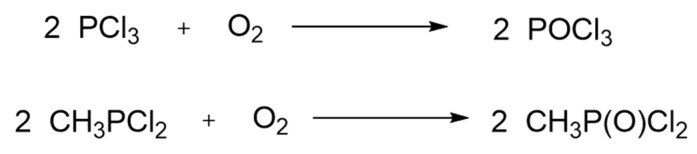
Some minor side reactions presented during the production of SW.

**Figure 13 ijms-24-08600-f013:**
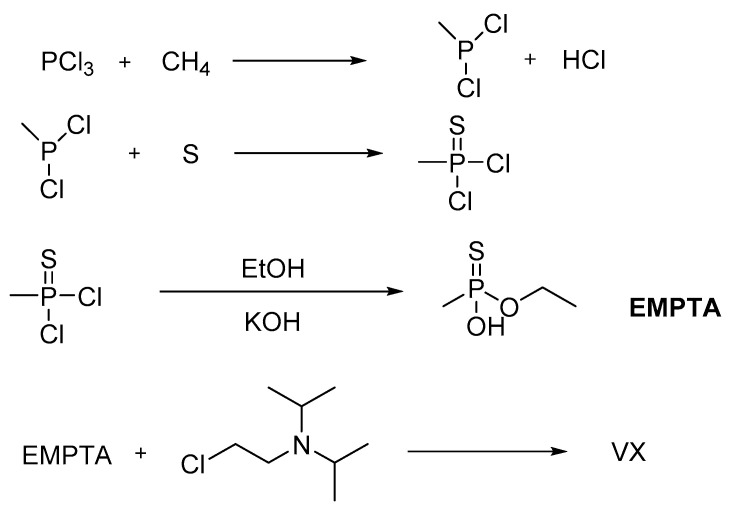
A schematic diagram of the EMPTA process to produce VX.

**Figure 14 ijms-24-08600-f014:**
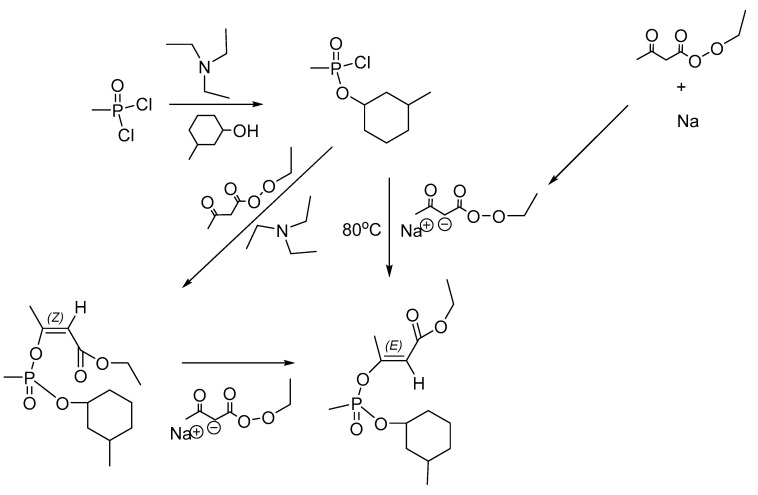
Synthesis of the *trans* EA-1576 agent.

**Figure 15 ijms-24-08600-f015:**
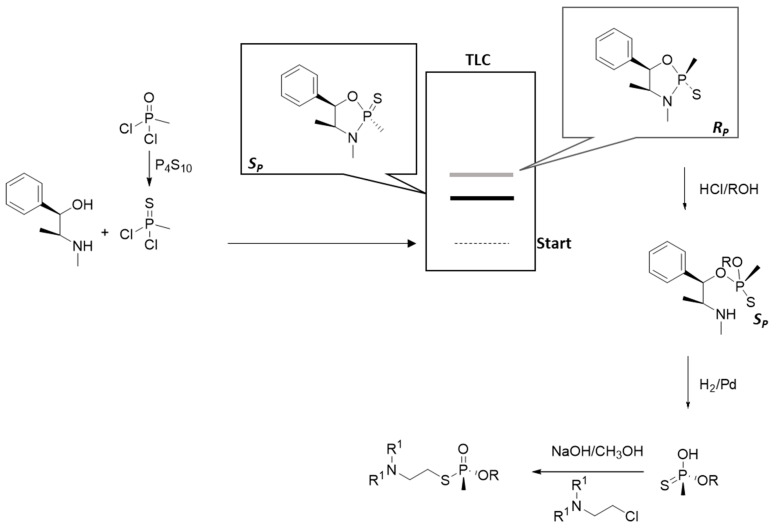
Synthesis of isolated enantiomers of V-agents.

**Figure 16 ijms-24-08600-f016:**
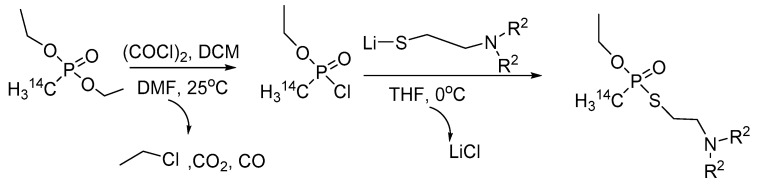
Synthesis of radiolabeled VX and VM. The lithium salt of N,N-dialkylaminoethanol can be prepared from the respectively substituted aminothiol with butyllithium.

**Figure 17 ijms-24-08600-f017:**
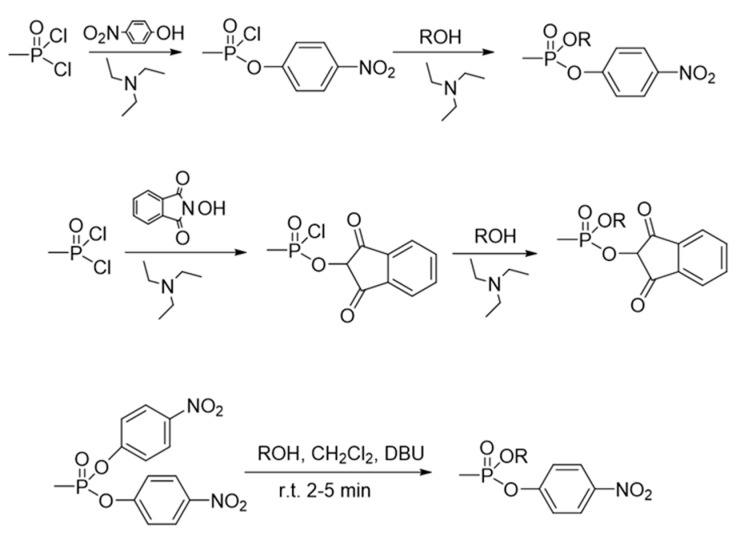
General synthesis schemes for V-agent surrogates.

**Figure 18 ijms-24-08600-f018:**
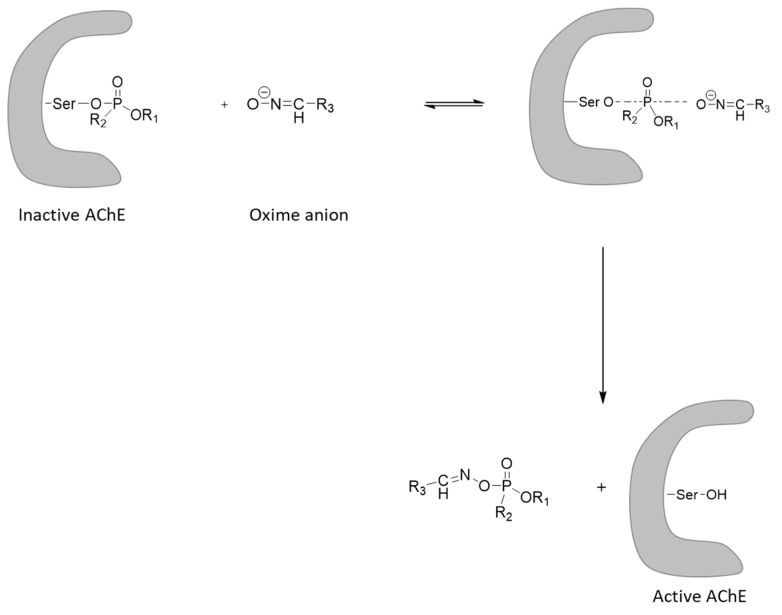
Potent mechanism of oxime reactivation of cholinesterase conjugate through nucleophilic attack.

**Figure 19 ijms-24-08600-f019:**
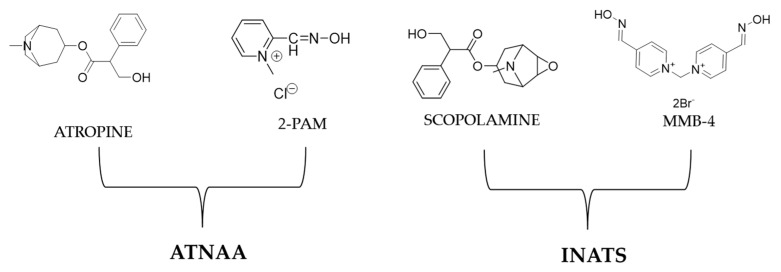
Chemical structures of the components of the ATNAA and INATS.

**Table 1 ijms-24-08600-t001:** Physical properties of V-agents. The data were obtained from [17] unless stated otherwise.

	@25 °C								
V-Agent	Density, g/mL	Viscosity, cS	Surface Tension, mN/m	Vapor Pressure, mmHg	Melting Point, °C	Flash Point, °C	Lower Consolute T in Water, °C	Formula	MW
VG (EA-1508)	1.0457	4.74	31.0		ca. −51		25.5	C_10_H_24_NO_3_PS	269.34
VP (EA-1511)	1.023	29.6	30.4					C_15_H_24_NO_3_P	297.33
EA-1576	1.0829	23.3	32.4					C_14_H_25_NO_5_P	304.32
EA-1521	0.995							C_16_H_34_NO_2_PS	335.49
EA-1622 ^#^	1.023	6.10	29.7			135		C_10_H_24_NO_2_PS	253.34
EA-1699 ^#^	1.0600	5.31	32.0					C_7_H_18_NO_2_PS	211.26
VM (EA-1664)	1.0311	5.85	31.2				74	C_9_H_22_NO_2_PS	239.32
VR (RVX, EA-4243)	1.0065	8.34 ^##^ 8.58	26.9 ^##^	0.00063 ^##^		150 ^###^		C_11_H_26_NO_2_PS	267.37
EA-1694 ^#^	1.0453	4.92	31.5				misc. 0–100	C_8_H_20_NO_2_PS	225.29
VE (EA-1517)	1.0180	5.44	29.5			157	41.4	C_10_H_24_NO_2_PS	253.34
VS (EA-1677)	1.0016	9.36	29.9		ca. −35	168	ca. −5	C_12_H_28_NO_2_PS	281.39
VX (EA-1701)	1.0083	9.96 10.10 ^#^ 10.09 ***	31.6 30.2 ^#^	0.000884 ^#^ 0.000878 *		159	9.4	C_11_H_26_NO_2_PS	267.37
EA-1728	0.9899	11.4	29.2		ca. −12	170	−1.6	C_12_H_28_NO_2_PS	281.39
EA-1763	0.9973	11.3	30.2				below 0	C_12_H_28_NO_2_PS	281.39
EA-3148	1.05 **	1.96 **		0.0004 **				C_12_H_26_NO_2_PS	279.38
EA-3317	1.02 **	35.1 **		0.00014 **				C_14_H_30_NO_2_PS	307.43
CVX (EA-6043)	1.0125 ***	9.29 ***	22.68 ***	0.00025 ***				C_11_H_26_NO_2_PS	267.37

^#^ indicates compounds with low purity; thus, the real values may differ from the reported. ^##^ [18], ^###^ [19], * [20], ** [21], *** [22].

**Table 2 ijms-24-08600-t002:** Toxicities of V-type nerve agents (μg·kg^−1^).

V-Agent	Mouse				Rat						Rabbit			Monkey	Cat	Guinea Pig	
	*i.v.*	*i.m.*	*p.o.*	*s.c.*	*i.v.*	*s.c.*	*p.c.*	*i.p.*	*i.m.*	*p.o.*	*i.v.*	*p.c.*	*s.c.*	*i.v.*	*i.v.*	*p.c.*	*s.c.*
VX	14.5 [25]	26.8 [25]	44.2 [25]	22 [26]	8.2 [25,26] 7 [27]	11.9 [25,26]	80 [25]	45.6 [25,26] 37 [27]	14 [25]	85 [25] 122 [26]			13.1 [28]	6 [27]	5 [27]	613 [29]	9 [29]
VR					14.5 [25] 15.3 [26]	15.9 [25,26]	290 [25]	63 [25,26]	14.1 [25]	20 [25]							11.3 [29]
VM									20 [25]	212 [25]						1289.9 [29]	14.9 [30]
VE (EA-1517)											15.3 [31]	40.7 [31]					
VP (EA-1511)											36.8 [31]	81.8 [31]					
EA-3148											3.1 [32]			4.5 [32]			
EA-1728							59.1 [33]		46 [25]	56 [25]							
EA-1694									28.9 [25]	121 [25]							
iPr-Me *									96 [25]	874 [25]							
EA-1699					17 [34]			54.5 [34]	23.6 [34]	121.9 [34]							
VG (EA-1508)				190 [35]		150 [35]					52.3 [31]	167.3 [31]	125 [35]				80 [35]
Seleno-VG				60 [12]													
Seleno-VE				21 [12]													

* CH_3_P(O)(OiPr)SCH_2_CH_2_N(CH_3_)_2_.

**Table 3 ijms-24-08600-t003:** Compounds detected during the assassination of Kim Jong-Nam.

	Detection		
Compound	1st Woman	2nd Woman	Victim
VX		x	x
iPr_2_NCH_2_CH_2_Cl		x	x
iPr_2_NCH_2_CH_2_SH		x	x
CH_3_P(O)(OEt)SH			x
CH_3_P(O)(OEt)OH	x		x
iPr_2_NCH_2_CH_2_SCH_2_CH_2_NiPr_2_			x
iPr_2_NCH_2_CH_2_SSCH_2_CH_2_NiPr_2_		x	x
(CH_3_)_2_NCH_2_CH_2_OH			x

## Data Availability

Not applicable.

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
