# Peer review of "Chemical, Physical, and Toxicological Properties of V-Agents"

_ijms, 2023, doi:10.3390/ijms24108600_

Round 1

Reviewer 1 Report

The review provides a holistic overview of V-agents, which are highly toxic organophosphate nerve agents, including subclasses such as phosphonylated thiocholines, phospho(n/r)ylated selenocholines, and non-sulfur containing agents like VP and EA 1576. The review also discusses their production, physical properties, toxicity, and stability during storage. V-agents pose a percutaneous hazard and can contaminate exposed areas for weeks, as demonstrated in the 1968 VX accident in Utah, and there is increasing concern about potential terrorist production and use, making it important to study their chemistry and develop countermeasures. The authors need to address the following concerns before the paper is considered to be published.

1. 8.5 Small scale synthesis for specific purposes:

Please add the solvent used in thin layer chromatography.

2. More references are needed. For example:

Line 121,

“The study of the V-agents’ physical properties is important since it correlates with their potential environmental spreading and contamination”

Page 10,

“Indeed, NTE is inhibited by certain organophosphate agents … weakness, ataxia, muscle twitching in the legs, and tingling.”

“It should be mentioned that rodents, although are widely used as animal models … for screening for OPIDN.”

“V-type nerve agents have not been used in military conflicts …the pathological effects of VX and EA-3148 have tested at low levels in human volunteers.”

Page 11,

“Class 1 V-agents are subjected to gradual deterioration … p-diethyldimethylpyrophosphonate (referred as pyro), ethyl methylphosphonic acid, alcohols and water.”

65 references should not be enough for an 18-page review. Please add more references accordingly. I only listed a few examples above. Some references were repeated in the review. It would be better to use different ones.

Author Response

Reviewer 1

The review provides a holistic overview of V-agents, which are highly toxic organophosphate nerve agents, including subclasses such as phosphonylated thiocholines, phospho(n/r)ylated selenocholines, and non-sulfur containing agents like VP and EA 1576. The review also discusses their production, physical properties, toxicity, and stability during storage. V-agents pose a percutaneous hazard and can contaminate exposed areas for weeks, as demonstrated in the 1968 VX accident in Utah, and there is increasing concern about potential terrorist production and use, making it important to study their chemistry and develop countermeasures. The authors need to address the following concerns before the paper is considered to be published.

Answer

We would like to thank the Reviewer for the constructive comments.

  1. 8.5 Small scale synthesis for specific purposes:

Please add the solvent used in thin layer chromatography.

Answer

It has been added as requested.

  1. More references are needed. For example:

Line 121, “The study of the V-agents’ physical properties is important since it correlates with their potential environmental spreading and contamination”

Page 10, “Indeed, NTE is inhibited by certain organophosphate agents … weakness, ataxia, muscle twitching in the legs, and tingling.”

“It should be mentioned that rodents, although are widely used as animal models … for screening for OPIDN.”

“V-type nerve agents have not been used in military conflicts …the pathological effects of VX and EA-3148 have tested at low levels in human volunteers.”

Page 11, “Class 1 V-agents are subjected to gradual deterioration … p-diethyldimethylpyrophosphonate (referred as pyro), ethyl methylphosphonic acid, alcohols and water.”

65 references should not be enough for an 18-page review. Please add more references accordingly. I only listed a few examples above. Some references were repeated in the review. It would be better to use different ones.

Answer

We have now significantly updated the reference list, and all aforementioned suggestions have been addressed.

Reviewer 2 Report

Dear Authors,

I found you article well structured and well written. You are right, that such article was not published before hence it makes it kind a unique. Certainly, you did good job. I recommend you article for publication after incorporating several minor revisions that have to be done throughout the manuscript.

1) What does the labeling "EA" means? It is the first time I see this as I always seen V-agent labes. I think it should be stated in manuscript as it is not common. Also, some time u used dash between EA and number. It should be unified.

2) I do not like your usage of units. You used "g kg-1" which is not correct, it should be "g.kg-1", it should be revised throughout the manuscript.

3) Page 2, line 67, there is "umole kg-1" it should be corrected to "umol.kg-1"

4) The units degree celsius (°C) are weird in all manuscript, the degree is somewhere in the middle of the letter C and not as it should be.

5) Chemical structures are not unified. Sometimes, those are simplified with just bonds between carbons (I recommend to use this in whole manuscript) and sometimes they are depicted as CH2CH2CH3 etc. It should be fixed, use simplified organic structures

6) In text, chemical formulas, e.g. CH2CH2CH3 have numbers not in subscript format, which is incorrect. Instead, there are somehow weird small numbers. It should be corrected in all manuscript.

7) Is there any other name/label for compound Vx? I found this designation very unfortunate as it could be very easily misplaced with VX.

8) Figure 8, should be revised. I found this figure disorganized and difficult to understand. If it is autocatalytic hydrolysis of V-agent why there is Et methylphosphonic acid reaction with VX? It means that VX is firstly hydrolyzed to Et methylphosphonic acid which further reacts with VX to form "pyro" compound? If yes it should be stated like that and also visualized like that. Also the bonds of chemical structure are incorrect, please clean all the structures in all manuscript.

9) Figure 9: why 1/8 S8 and not just sulfur or S? This is kind a weird expression. By the way, same issue as already mentioned in 5)

10) Table 3, again, as mentioned in 6)

11) Figure 10, what does the "[O]" means? Oxidation? Presence of oxygen? Use at least term "oxidation" or O2

12) Figure 11, Since is it major industrial production process, I suggest to show whole process in this figure and not just this part that is shown

13) page 16, first sentence of second paragraph, there is typo "there is can be"

14) O-ethyl etc. "O-" should be italic type, fix it in all manuscript

15) Figure 13, clean the structures, in general, clean all the structures in whole manuscript

16) Page 18, abbreviation CAS for chemical attribution signature is unfortunate and should be avoided. CAS is generally accepted for chemical abstract number hence it is misleading. Besides, such abbreviation is used only once and therefore is has no sense to be abbreviated.

17) Conclusion should be probably revised, although, I understand the final message and agree with their conclusion. Try to better conlude what is this manuscript about and what are its benefits. The middle four lines sentence should be divided. 

Author Response

Reviewer 2

Dear Authors,

I found you article well structured and well written. You are right, that such article was not published before hence it makes it kind a unique. Certainly, you did good job. I recommend you article for publication after incorporating several minor revisions that have to be done throughout the manuscript.

Answer

We would like to thank the Reviewer for the nice comments and the helpful suggestions.

  • What does the labeling "EA" means? It is the first time I see this as I always seen V-agent labes. I think it should be stated in manuscript as it is not common. Also, some time u used dash between EA and number. It should be unified.

Answer

EA is abbreviation of Edgewood arsenal.  It has been added in the text. In the revised version, all designations are EA with dash followed by number.

  • I do not like your usage of units. You used "g kg-1" which is not correct, it should be "g.kg-1", it should be revised throughout the manuscript.

Answer

It has been revised accordingly.

  • Page 2, line 67, there is "umole kg-1" it should be corrected to "umol.kg-1"

Answer

It has been revised accordingly.

  • The units degree celsius (°C) are weird in all manuscript, the degree is somewhere in the middle of the letter C and not as it should be.

Answer

It has been revised accordingly.

  • Chemical structures are not unified. Sometimes, those are simplified with just bonds between carbons (I recommend to use this in whole manuscript) and sometimes they are depicted as CH2CH2CH3 It should be fixed, use simplified organic structures.

Answer

The figures have been revised and now all formulas are shown in the same format.

  • In text, chemical formulas, e.g. CH2CH2CH3have numbers not in subscript format, which is incorrect. Instead, there are somehow weird small numbers. It should be corrected in all manuscript.

Answer

They have been revised accordingly.

  • Is there any other name/label for compound Vx? I found this designation very unfortunate as it could be very easily misplaced with VX.

Answer

Vx has been revised to V-sub x to avoid confusion with VX.

  • Figure 8, should be revised. I found this figure disorganized and difficult to understand. If it is autocatalytic hydrolysis of V-agent why there is Et methylphosphonic acid reaction with VX? It means that VX is firstly hydrolyzed to Et methylphosphonic acid which further reacts with VX to form "pyro" compound? If yes it should be stated like that and also visualized like that. Also the bonds of chemical structure are incorrect, please clean all the structures in all manuscript.

Answer

First, the "pyro" compound is hydrolyzed and the hydrolysis product reacts with VX. In Figure 8, the initial step includes the reaction of pyro with water and is shown in dashed box. Further, next to pyro the term byproduct has been added.

9) Figure 9: why 1/8 S8 and not just sulfur or S? This is kind a weird expression. By the way, same issue as already mentioned in 5)

Answer

It has been revised accordingly.

10) Table 3, again, as mentioned in 6)

Answer

They have been revised accordingly.

11) Figure 10, what does the "[O]" means? Oxidation? Presence of oxygen? Use at least term "oxidation" or O2

Answer

We have replaced the [O] with oxidation as suggested.

12) Figure 11, Since is it major industrial production process, I suggest to show whole process in this figure and not just this part that is shown

Answer

It has been revised accordingly. A new Figure has been added.

13) page 16, first sentence of second paragraph, there is typo "there is can be"

Answer

It has been revised accordingly.

14) O-ethyl etc. "O-" should be italic type, fix it in all manuscript

Answer

It has been revised accordingly.

15) Figure 13, clean the structures, in general, clean all the structures in whole manuscript

Answer

It has been revised accordingly.

16) Page 18, abbreviation CAS for chemical attribution signature is unfortunate and should be avoided. CAS is generally accepted for chemical abstract number hence it is misleading. Besides, such abbreviation is used only once and therefore is has no sense to be abbreviated.

Answer

It has been revised accordingly.

17) Conclusion should be probably revised, although, I understand the final message and agree with their conclusion. Try to better conlude what is this manuscript about and what are its benefits. The middle four lines sentence should be divided. 

Answer

The conclusions have been revised and the four-line sentence has been divided.